# Changes in Body Composition and Physical Performance in Children with Excessive Body Weight Participating in an Integrated Weight-Loss Programme

**DOI:** 10.3390/nu14173647

**Published:** 2022-09-03

**Authors:** Magdalena Dettlaff-Dunowska, Michał Brzeziński, Agnieszka Zagierska, Anna Borkowska, Maciej Zagierski, Agnieszka Szlagatys-Sidorkiewicz

**Affiliations:** Department of Paediatrics, Gastroenterology, Allergology and Paediatric Nutrition, Medical University of Gdańsk, 80-803 Gdańsk, Poland

**Keywords:** obesity, weight–loss, body composition

## Abstract

The problem of overweight and obesity is a growing phenomenon in the entire population. Obesity is associated with many different metabolic disorders and is directly associated with an increased risk of death. The aim of the study was to assess the changes in body composition and physical fitness in children participating in an integrated weight-loss programme and to analyse the possible relationship between changes in body composition and improvements in fitness. Participants of the study were recruited from the “6–10–14 for Health”-multidisciplinary intervention programme for children aged 6 to 15 years old. A total of 170 patients qualified for the study, and 152 patients were enrolled. Statistically significant changes in body composition were found after the end of the intervention program, as measured by both BIA (bioimpedance) and DXA (Dual Energy X-ray Absorptiometry). The differences in KPRT (Kasch Pulse Recovery Test) results at baseline and after intervention are positively correlated with the difference in fat mass between baseline and the after-intervention measure. Improving physical fitness is positively correlated with a decrease in FM (fat mass) and an increase in FFM (fat-free mass) measured in both absolute values and %. Both BIA and DXA methods proved to be equally useful for measuring body composition.

## 1. Introduction

The burden of overweight and obesity is a growing phenomenon in the entire world population [1]. This worrying situation is observed worldwide, with a higher prevalence in lower socio-economic and educational strata [2,3]. Obesity is associated with many different metabolic disorders such as insulin resistance, symptomatic diabetes, lipid disorders and cardiovascular disorders [4,5,6]. Obesity is directly associated with an increased risk of death. Due to the multitude of complications associated with childhood obesity, early implementation of intervention programmes is extremely important. To date, analyses indicate the predominance of interdisciplinary programmes covering the entire family [7]. To date, data on surgical and pharmacological interventions in children are inadequate [8,9,10]. However, quite a lot of analyses were carried out on the impact of interventions related to lifestyle, dietary supplementation (green tea, yerba mate) and DHA (docosahexaenoic acid) [11,12,13,14]. The influence of vitamin D on body-mass reduction in children and adults was also investigated [15,16]. Fat reduction is associated with the normalisation of metabolic parameters, such as inflammation markers, lipid profile, insulin resistance and blood pressure [5,6,17,18]. It was also noted that weight loss on its own seems to be insufficient for improving metabolic parameters and physical fitness, with the improvement in body composition, understood as fat loss and increase in muscle mass, being more important. That is why early and effective intervention increases the likelihood of maintaining the health potential developed through such interventions. 

A clinically important problem faced when conducting treatment aiming to reduce excess body weight in children is the selection of a method to assess the effectiveness of treatment. The paediatric population is particularly challenging due to the nature of childhood as a period of growth. On the one hand, we want to reduce body weight, but weight loss therapy cannot lead to nutritional deficiencies, body composition disorders and, consequently, developmental disorders. DXA remains the gold standard for body composition assessment; however, it is quite difficult to access and onerous due to the need to use X-rays. BIA (electrical bioimpedance test) is more accessible and less burdensome for the patient. A good correlation between both methods for assessing body composition has been reported in several studies [19,20]. 

Unfortunately, the prevalence of overweight and obesity in children in Poland is not subject to regular epidemiological studies. The data published so far often cover different age groups and different diagnosis criteria, which makes comparisons significantly difficult [21,22]. Analysing the local data of 12,330 children from 2008–2016 years of children from Gdańsk, the prevalence of overweight among children aged 6–7 remains at a low level, remains stable over the past 20 years, and is higher in girls [23]. The prevalence of metabolic syndrome in the studied population was 12.9%—similar to other populations [24].

The primary aim of this study was to assess changes in body composition and physical fitness in children with vitamin D deficiency during a weight reduction program. Moreover, we wanted to analyse whether the changes in body composition (FM, FFM) are better indicators of physical performance improvement than BMI in order to propose reliable monitoring practices during weight reduction programs.

## 2. Materials and Methods

All participants of the study were recruited from the “6–10–14 for Health” programme run by the University Clinical Centre in Gdańsk. It is a multidisciplinary intervention programme aimed at children aged 6 to 15 years old and their parents. As part of the project, children aged 6 to 15 years old attending schools in Gdańsk are examined every 3–4 years by special teams (paediatricians and/or nurses). All children with a BMI exceeding the 85th percentile are invited to participate in the “6–10–14 for Health “. Participants in the “6–10–14 for Health” programme and their family members (parents/carers) receive an integrated intervention over a period of 12 months, including individual medical care (paediatrician), dietary advice and physical and psychological counselling during one integrated session (4 × 20–25 min). This intervention is offered to all programme participants for a period of 12 months, with check-up visits every 3 months. Details of the study protocol, participants and intervention plan had been published in advance [25,26].

### 2.1. Study Design

The primary study, on the basis of which the current “post hoc” analysis is conducted, is planned for the assessment of the impact of vitamin D supplementation on the change in body weight and its composition. The results of the analysis conducted by Brzeziński [26] did not confirm the effect of supplementation on the improvement in weight reduction, and no differences in body composition (fat mass, fat-free mass measured by both BIA and DXA) between the group with vitamin D supplementation and the placebo group were found. Therefore, the analysis reported in this study includes a unified group of patients with vitamin D deficiency (<30 ng/mL vitamin D level) at the time of joining the study. 

### 2.2. Participants

Participants were children 6, 9–11 and 14 years old, according to the principles of the “6–10–14 for Health” programme, as follows:

Inclusion criteria: overweight BMI ≥ 85th < 95th percentile) or obesity (BMI ≥ 95th percentile), identified on the basis of anthropometric parameters using the Polish reference centile charts—OLAF project [22]; blood concentration 25 (OH) D3 < 30 ng / ml; written informed consent of legal guardians.

Exclusion criteria: Chronic conditions (asthma or allergies, inflammatory diseases of the connective tissue, gastrointestinal disorders, kidney and liver disorders, bone metabolism disorders); contraindications to the administration of vitamin D; administration of any preparation containing vitamin D, calcium or steroid hormones in three months prior to the study enrolment.

A total of 170 patients qualified for the study, and 152 patients were enrolled. A total of 85 patients were included in the study group and 67 in the placebo group. The study was completed by 56 participants in the study group and 53 in the placebo group. 

The pattern of participation in the study was as follows.

### 2.3. Methods 

Kasch Pulse Recovery Test (3 min step-test) was used to assess physical fitness. It consists of stepping up and down for 3 min onto a 30 cm high step and measuring pulse before and after the exercise. It is a modification of the Harvard test. The advantage of the test, which makes it very useful in population studies, is that it does not cause nervous system fatigue. Previous studies have shown that children of early school age have rapid tiredness with prolonged monotonous effort [27]. 

Bioelectrical Impedance Analysis (BIA) is a body composition analysis method that uses very low current. This test takes advantage of the fact that individual tissues in the body contain more or less water and therefore conduct electricity to varying degrees. The TANITA DC-430 S MA device was used for measurements.

Dual Energy X-ray Absorptiometry (DXA). Method for the evaluation of mineral bone, muscle and fat density using X-rays. In paediatric patients, it is recommended to analyse data on total bone mass without division into components. 

### 2.4. Bioethics Committee Approval and Consent to Participate

The study was approved by the Independent Bioethics Committee for the University of Gdańsk, Poland, [NKBBN/130-206/2015] of 25 May 2015. Parents/guardians gave their written consent prior to the commencement of any testing procedure. The study protocol was reviewed by an independent financial committee (Nutricia Foundation) during the process of applying for funding.

Study registration number: NCT 02828228; date of registration for the study: registered in ClinicalTrials.gov on 8 June 2016.

### 2.5. Study Procedure

The analysis included results of measurements made during the appointments completed within the obesity treatment programme “6–10–14 for Health” using the 0, 3, 6, and 12 months schedule. All visits included individual meetings with a paediatrician, nutritionist, physical activity specialist and psychologist. Detailed information about the programme can be found in a previous publication. At the first visit, all children’s carers were asked to give their written consent to the child’s participation in the study. The refusal to participate in the study did not affect the participation in the intervention programme. Children with low 25 (OH) D3 (<30 ng/mL) were referred to DXA (dual-energy X-ray absorption) within two weeks of the initial visit. Enrolled participants were randomly assigned to one of the two groups using a computer-generated randomisation table. The participants were then randomly assigned to one of two test groups:

GROUP I (vitamin D group): medical intervention, dietician, psychologist and physical education specialist, parent education + oral administration of vitamin D3 (1200 i.u. daily) for 26 weeks. 

GROUP II (placebo group): medical intervention, dietician, psychologist and physical education specialist, parent education + daily oral placebo administration for 26 weeks.

### 2.6. Randomisation and Blinding

The randomisation list was generated by the Office of Clinical and Scientific Research, University Clinical Centre (OCSR UCK), which did not clinically participate in the study, using a computer program (StatsDirect) with an allocation ratio of 1:1 and block size of 6. The allocation sequence was concealed from the researchers responsible for registering and evaluating participants. Throughout the study, all investigators, participants, outcome evaluators and data analysts were blinded. The allocation to the groups and distribution of the drug/placebo was carried out by an independent researcher (MS-F) who was not directly involved in the intervention program.

### 2.7. Dispensing Treatment and Compliance Assessment

Both treatments (vitamin D (1200 IU) and placebo) were provided by the company (Sequoia) in identical capsules and packs (5 capsules per blister, 6 blisters per box). Sets of 7 boxes (6 months’ worth of treatment) were prepared and blinded by the Office of Clinical and Scientific Research, University Clinical Centre. Blister packs and boxes were collected from the patient during the last visit. The number of remaining capsules was documented to enable compliance assessment. 

### 2.8. Outcome Measures

Change in the BMI percentile, change in body composition and improvement in physical fitness after the end of the intervention programme.

### 2.9. Sample Size

Assuming a probability of the event (at least a 10% reduction in BMI during the observation period) at 0.85 and 0.6 for the experimental and control groups, respectively, the minimum sample size was estimated to be 130 (65 per group) to achieve the power of 0.9 for alpha equal to or less than 0.05 and beta equal to or less than 0.1.

### 2.10. Statistical Analysis

The normal distribution of continuous variables was verified with the Shapiro–Wilk test. Descriptive statistics are presented as the mean or median and standard deviation from the mean. Between-group comparisons were carried out using the Mann–Whitney U test, and in-group comparisons used the Wilcoxon test. Nonparametric tests were chosen because of the large number of significant Shapiro tests, which were used for normality assumption assessment. All statistical tests were 2-tailed and performed at the 5% level of significance. All analyses were performed on the intention-to-treat basis, in which all of the participants in a trial are analysed according to the intervention to which they were assigned, analysing only the participants who completed the whole weight management intervention programme. Statistical analyses were performed with Statistica 13 (TIBCO Software Inc., Tulsa, OK, USA (2017)).

## 3. Results

Descriptive information about the study participants is presented in Table 1.

The study enrolled 152 participants, and 109 children completed the program. The average age of participants was 10.93 years old, and the age range at the start was 5.8 to 15.36 years old. The average increase in participants’ height was 5.36 cm. The average difference in BMI percentiles was −3.43, but it is worth noting the range of results from −39 to 10 (Table 1). 

Attention is drawn to the variable number of respondents in individual tables. It results from participants dropping out at different stages of the programme and from the inability to obtain full results in some cases.

The results demonstrate no significant correlation between the initial BMI percentile and vitamin D level at the first visit. Additionally, the correlation did not show a significant impact of the change in BMI on changes in vitamin D levels after the supplementation period. Negative correlations (*p* < 0.05) between % body fat and vitamin D levels were noted at both visits—which may confirm the relationship between body fat and blood vitamin D levels. However, these correlations were weak (−0.25 to −0.21) and not confirmed in the DXA tests.

When analysing the data on the effect of supplementation with 1200IU of vitamin D, no effect on the improvement of physical fitness in children with vitamin D deficiency was observed. Therefore, further analyses were carried out after group unification (Table 2).

Statistically significant changes in body composition were found after the end of the intervention program, as measured by both BIA and DXA.

The analysis of body composition using the electric bioimpedance method included 106 participants. The average loss of adipose tissue mass was 0.05 kg, which corresponds to −1.7% of the body. The average increase in muscle mass was 2.89 kg, which corresponds to an increase of 1.43%.

There was a statistically significant (*p* < 0.0) decrease in almost all measured bioimpedance parameters between visits one and four, but not for adipose tissue mass (Table 3).

DXA analysis included 95 patients. A fat mass loss of 1.28% on average was demonstrated, resulting in an average loss of 704 g. Similarly, the gain in lean mass was 1.28% on average. When analysing the change in total lean, an increase of 1.28% was shown, which gives an increase of 2886.68 g on average. The difference between values on the first and last visit was statistically significant. 

Furthermore, we measured the impact of physical activity level - measured with a simple step test on body composition parameters. There were no significant correlations between the level of physical activity on first, last or the difference of both and the achieved difference in body composition parameters (Table 4). 

The differences in KPRT results at baseline and after intervention are positively correlated with the difference in fat mass (5) between baseline and the after-intervention measure (Table 5).

In the analysis of the relationship between the change in body composition and physical fitness, a statistically significant positive correlation between the improvement in the percentage of muscle mass in relation to the improvement in the KPRT score was found in the BIA assessment (0.488293). A correlation was found between the percentage decrease in fat mass and the improvement in physical fitness.

In the DXA assessment, the p-value for fat mass loss was −0.411392. In addition, the p-value for the increase in lean body mass was 0.411392, which is interpreted as a strong confirmation of statistical significance and, thus, of the impact on the improvement in KPRT.

No statistically significant relationships were observed between the improvement in bone parameters (assessed as SPINE BMD and SUBTOTAL BMD) and the improvement in physical fitness.

## 4. Discussion

Intervention programmes for the treatment of childhood obesity should aim to monitor its biological effects. The main task of a programme for the treatment of childhood obesity is not only to reduce body weight or BMI but, above all, to improve body composition (decrease adipose tissue and increase muscle mass) and, consequently, improve physical fitness. 

Literature analysis of the topic shows how important it is to assess body composition and not only BMI in relation to percentile charts. Even in childhood, the excess of adipose tissue, especially visceral adipose tissue, is related to an elevated risk of metabolic diseases such as insulin resistance, hyperinsulinemia and impaired glucose tolerance, abnormal fasting glucose levels, symptomatic diabetes, lipid and cardiovascular disorders and hypertension [5,6,17,18]. Numerous studies report on the long-term effects of obesity that occur in childhood [28]. Obese children are even as much as five times more likely to be obese at a later age, and as many as 80% of obese adolescents will be obese as adults [29,30]. However, it should be remembered that obesity in childhood is not the only determinant of obesity among adults because as many as 70% of obese adults were not obese in childhood [31]. 

Several ways of assessing the effectiveness of interventions are used in scientific research and studies on the assessment of weight loss methods. The most commonly used are weight assessment, BMI and body composition analysis. Researchers agree that since adipose tissue is the most metabolically harmful type of tissue, effective evaluation of the effectiveness of reduction programs should primarily take into account the measurement of % adipose tissue and not only BMI or body weight [31].

In our study, we included measurements not only of body weight and BMI but also body composition and also analysed the improvements in physical fitness. 

Health benefits of weight reduction programmes consisting of dietary intervention and exercise in children and adolescents with obesity are reported to benefit both metabolically healthy participants (MHO) and metabolically unhealthy (MUO) [17]. This publication assessed a large group of children from Shanghai under an intervention programme. Importantly, the researchers emphasise that the metabolically healthy state is temporary and may change as early as childhood or during adolescence. The study showed the benefits of diet and exercise-based interventions to improve body weight, BMI, % adipose tissue, waist circumference, total cholesterol, and LDL cholesterol, while SBP, DBP, RHR, FBG, and TG levels improved more markedly in the MUO group than in the MHO group. The childhood period has important implications for maintaining adult metabolic health, and early interventions and strategies to lower blood pressure and prevent a decline in HDL-C levels can contribute to maintaining and developing the health potential of the population. Similar conclusions emerge from the studies involving adults [32]. Particularly worth emphasising is the fact that only a long-term approach aimed at a sustainable lifestyle change can improve metabolic profiles, which is confirmed by interventional studies in adults [33].

More and more reports appear confirming the positive importance of a multimodal approach to intervention programs and the use of modern technologies such as smartphone applications [34]. In this study, the intervention that featured an educational game on a smartphone showed more promising results in terms of weight loss, waist circumference, hip circumference and lipid profile. In addition, it was shown that the psychological assessment of the attitude towards the intervention was better. Chinese research [35] confirms the importance of a multidirectional and technology-based approach to creating programs supporting weight loss, where higher BMI percentile at baseline and increased frequency of mobile app usage were directly related to more significant weight loss. Another confirmation of the positive impact of programs combining the benefits of dietary modification and physical activity is that the study showed an improvement in the parameters of myocardial efficiency and a reduction in the thickness of the inner and middle carotid artery membranes in patients showing a reduction in systemic blood pressure in obese children [36]. Moreover, patients under frequent medical check-ups have a lower risk of failure of weight reduction [37].

Many studies confirm the impact of the family environment on children’s BMI [38,39,40,41].

Reports indicate a strong positive relationship between an adult’s BMI change and a child’s BMI change, and thus the active involvement of adult family members in the weight loss process improves children’s treatment outcomes. In our study, the entire family was also provided with care and education, but the measurements concerned only the participant children. 

Lifestyle and physical activity are key elements in maintaining a high health potential. Based on the analysis published in 2016, it was found that physical activity, regardless of the degree of intensity, has a positive impact on physical, psychological/social and cognitive health indicators. The analysis confirmed that at least 60 min of moderate or vigorous activity daily is beneficial for the cardiovascular system in the paediatric population [42].

Numerous programs and interventions confirm the beneficial effect of regular and organised physical exercise on improving metabolic parameters and cardiovascular system efficiency. In an intervention consisting of a low-calorie diet and vigorous physical exercise conducted by Chinese researchers, a reduction in BMI by 3.3 kg/m^2^ and fat by 4% was achieved [43]. In our studies, the decrease in % body fat was 1.7% and 1.28% using BIA and DXA, respectively. Another study on the intervention consisting of the introduction of additional sports activities in the form of football training also confirmed the beneficial effect of physical effort on improving body composition [44].

An important element in monitoring the effectiveness of obesity treatment is the ability to easily and quickly assess cardiovascular and physical fitness. The KPRT method used in this study meets these criteria and is well tolerated by children of school age [27].

The fat loss we observed was correlated with better KPRT results. This confirms the assumptions of this work and offers important evidence in the discussion about the need to monitor not only the absolute value of the BMI but also the body composition. In children as young as five years old, training of moderate intensity results in an improvement in body composition parameters [45].

Due to the short observation period, we are not able to draw unequivocal conclusions about the extent of the impact of physical activity on bone tissue; in addition, children who were diagnosed with vitamin D deficiency participated in the study, which resulted in additional restrictions. Reports on the state of the skeletal system in children with obesity are inconclusive, which is confirmed by the analysis of the available literature carried out by Italian researchers in 2020 [46]. Undoubtedly, these aspects require further research.

In conclusion, physically fit children are more willing and easier to undertake physical activity as a form of spending free time. Such a model of spending time gives them a chance to maintain long-term benefits in adulthood and thus increase health potential in adulthood. Therefore, in order to actually assess the biological effects of weight loss programmes in children, we suggest using the bioimpedance method as a more accessible and less invasive method. The assessment of body composition as a measure of the result of the weight loss programme also allows us to focus not only on reducing body weight as the most important outcome but also on improving its quality. 

## 5. Conclusions

Improving physical fitness is positively correlated with a decrease in FM and an increase in FFM measured in both absolute values and %. Both BIA and DXA methods proved to be equally useful for measuring body composition. 

Since BIA and DXA have proved equally suitable for monitoring body composition, BIA seems to be a better method in clinical practice because of its greater accessibility and lower onerousness. 

Intervention programs should focus not only on a direct reduction of BMI but, above all, on improving body composition and lifestyle modification because only such action gives a chance to preserve long-term effects in the form of preserving and improving health capital.

## Figures and Tables

**Table 1 nutrients-14-03647-t001:** Descriptive statistics for the whole studied group.

	No of Patients	Mean ± SD	(95% CI)
Age v1 (years)	149	10.93 ± 2.97	10.45–11.41
Body mass v1 (kg)	152	58.08 ± 20.58	54.78–61.38
Body mass v4 (kg)	109	60.88 ± 18.76	57.32–64.44
Height v1 (cm)	152	150.14 ± 17.37	147.36–152.92
Height v4 (cm)	109	155.50 ± 16.16	152.43–158.57
BMI v1	152	24.78 ± 3.88	24.16–25.40
BMI v4	109	24.50 ± 3.72	23.80–25.21
Δ BMI v4–1	109	−0.18 ± 1.83	−0.53–0.16
BMI centile v1	152	95.21 ± 3.29	94.68–95.73
BMI centile v4	109	91.75 ± 8.55	90.13–93.37
Δ BMI centile v4–1	109	−3.43 ± 7.41	−4.84–(−2.02)

SD—standard deviation; CI—confidence interval; BMI—body mass index; v1—visit 1; v4—visit 4; Δ—delta–difference.

**Table 2 nutrients-14-03647-t002:** Analysis of changes in physical fitness in the studied groups after the end of the intervention program.

	No Patients (Vit D)	No Patients (Placebo)	Sum.Rank (Vit D)	Sum.Rank (Placebo)	*p* *
KPRT_rank_v1	55	58	2,995,500	3,445,500	0.404121
KPRT_rank_v4	19	23	381,000	522,000	0.479873
Δ KPRT rank v4–v1	19	22	430,000	431,000	0.412217

* U Mann–Whitney test; KPRT_rank-test value; v1—visit 1; v4—visit 4; Δ—delta–difference.

**Table 3 nutrients-14-03647-t003:** BIA results.

	No	Mean ± SD	(95% CI)
BI_FM (kg) visit 1	105	18.16 ± 7.66	16.83–19.35
BI_FM (kg) visit 4	105	18.21 ± 7.61	16.76–19.66
Δ BI_FM (kg) visit 4–1	105	−0.05 ± 4.03	−0.83–0.73 *
BI_FM (%) visit 1	105	31.19 ± 5.38	30.14–32.23
BI_FM (%) visit 4	105	29.48 ± 6.46	28.23–30.73
Δ BI_FM (%) visit 4–1	105	−1.70 ± 4.28	−2.53–(−0.87) #
BI_MM (kg) visit 1	106	37.53 ± 13.47	35.61–40.13
BI_MM (kg) visit 4	106	40.41 ± 12.88	38.02–42.89
Δ BI_MM (kg) visit 4–1	106	2.89 ± 2.69	2.36–3.39#
BI_MM (%) visit 1	106	65.37 ± 4.89	64.74–66.45
BI_MM (%) visit 4	106	66.69 ± 6.10	65.57–67.87
Δ BI_MM (%) visit 4–1	106	1.43 ± 3.87	0.70–2.19 #

* Wilcoxon pair test *p* = 0.57; # Wilcoxon pair test *p* < 0.05; Sp—spine, BMD—bone mineral density, Δ—delta–difference, TBLH—total body less head, TFM—total fat mass, TLM—total lean mass.

**Table 4 nutrients-14-03647-t004:** DXA Results.

	No	Mean ± SD		*p* *
Sp BMD visit 1	95	0.76 ± 0.18	0.72–0.78	0.000
Sp BMD visit 4	94	0.81 ± 0.19	0.77–0.85	
Δ in Sp BMD visit 4–1	94	0.05 ± 0.04	0.05–0.06	
TBLH BMD visit 1	95	0.86 ± 0.14	0.84–0.89	0.000
TBLH BMD visit 4	95	0.90 ± 0.14	0.87–0.93	
\Δ in TBLH BMD visit 4–1	95	0.04 ± 0.03	0.04–0.05	
TFM (kg) visit 1	95	24.4 ± 8.8	23.3–26.3	0.015
TFM (kg) visit 4	95	25.1 ± 8.8	23.3–26.9	
Δ in TFM visit 4–1	95	7.0 ± 4.7	−2.6–1.7	
TLM (kg) visit 1	95	31.6 ± 10.6	30.4–34.1	0.000
TLM (kg) visit 4	95	34.4 ± 11.2	32.2–36.7	
TFM (%) visit 1	95	43.38 ± 4.11	42.54–44.22	0.004
TFM (%) visit 4	95	42.10 ± 5.37	41.01–43.20	
Δ in TFM (%) visit 4–1	95	−1.28 ±4.04	−2.10–(−0.46)	
TLM (%) visit 1	95	56.62 ± 4.11	55.78–57.46	000.4
TLM (%) visit 4	95	57.90 ± 5.37	56.80–58.99	
Δ in TLM (%) visit 4–1	95	1.28 ± 4.04	0.46–2.10	

* Wilcoxon pair test; Sp—spine, BMD—bone mineral density, Δ—delta–difference, TBLH—total body less head, TFM—total fat mass, TLM—total lean mass.

**Table 5 nutrients-14-03647-t005:** Analysis of the correlation between differences in body composition and change in physical fitness.

Variable	KPRT		
	KPRT_1	KPRT_4	KPRT_4–1
Δ BI_FM (%) visit 4–1	0.173286	−0.191248	−0.542303
Δ BI_MM (%) visit 4–1	−0.170405	0.146202	0.488293
Δ BI_NFM (%) visit 4–1	−0.156313	0.149330	0.478494
Δ Sp BMD visit 4–1	−0.149281	0.171923	0.389636
Δ Sub BMD visit 4–1	0.069039	0.072368	−0.056962
Δ TFM (%) visit 4–1	0.082971	−0.279475	0.411392
Δ TLM (%) visit 4–1	−0.082971	0.279475	0.411392
Δ BMI % visit 4–1	−0.043510	−0.106357	−0.282023

Spearman rank correlation coefficient; bolded when *p* < 0.05; KPRT—Kasch Pulse Recovery Test; BI—bioimpedance; FM—fat mass; Δ—delta–difference; MM—muscle mass; BMD—bone mineral density; BMI—body mass index; NFM—non-fat mass; Sp—spine; Sub-subtotal; TFM—total fat mass; TLM—total lean mass.

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
