# Peer review of "Changes in Body Composition and Physical Performance in Children with Excessive Body Weight Participating in an Integrated Weight-Loss Programme"

_nutrients, 2022, doi:10.3390/nu14173647_

Round 1

Reviewer 1 Report

This is a welcome report on a weight loss programme in Poland. The research is well designed; the sample is good; and the findings will be of interest to the readers of this journal. To get the paper ready for publication, it is suggested that the authors give the following points some further thought and elaboration.

1. The paper begins, in the first sentence, by referring to "the entire country", without first mentioning the country. One would assume Poland, but it is better that the authors actually say that. In addition, it would be good if a little more background is given at the beginning of the situation regarding obesity and related metabolic conditions of children in Poland over the past few decades.

2. In the methodology section, it would be better if some more details are given about the research design, particularly with regards to the weight reduction programme.

3. On page 8 the authors mention the "multimodal approach". It would be helpful if this can be give a little elaboration and some relevant citations.

4. There are a number of grammatical and other typos throughout the manuscript, which I have marked along the margin.

Author Response

Thank you  for all your comments and suggestions. I would like to answer the issues raised.

  1. In the introduction, I added a paragraph relating to the situation in Poland and quite locally in Gdańsk, where this program was and still is. I also referred to the work describing the severity of the obesity problem over the past 20 years, published in 2020 by our team.
  2. Thank you for your suggestions, but this action was intentional, we did not want to duplicate the information that was described in great detail and published in the quoted publication.
  3. As suggested, I add a few more studies covering this area
  4. Of course, I took into account the corrections.

Reviewer 2 Report

In this manuscript (ID#: nutrients-1883505), entitled “Changes in body composition and physical performance in children with excessive body weight participating in an integrated weight-loss programme”, the authors, Dettlaff-Dunowska et al, assessed the changes in body composition and physical fitness in children participating in an integrated weight-loss program. Their study demonstrated that the intervention program significantly improved the body composition. However, this study is superficial, barely contributes to this research area. Several major concerns are listed in the following paragraphs:

1. This study observed the effect of a physical fitness program on body composition, which only evaluated the effectiveness of fitness program, without observing the beneficial effect on the obesity pathophysiology, such as high blood pressure, hyperlipidemia, hyperglycemia, and other metabolic change.

2. There is no significant precise conclusion in the current study. What is the aim of this research?

3. Vitamin D is mentioned in the manuscript (page 2 lines 79-86). What is the conclusion regarding the effect of vitamin D supplementation on the body mass and composition? Was endogenous vitamin D considered?

4. The manuscript is a research article, not a review article. Therefore, please focus on the new data collected in the current observation instead of discussion of others’ data.

5. What is “FFM”. Please define those abbreviations in the abstract. 

Author Response

Thank you  for all your comments and suggestions. I would like to answer the issues raised.

  1. The current study is a part of larger project analysing the effects of intergratred weiight-loss programme in obese children. The analysis of metabolic results are already presented in other papers [23,24,25, 26]. In intention of this paper was to present the changes in body composition and physical performance.
  2. I claryfied the aim of the study: The primary aim of this study was to assess changes in body composition and physical fitness in children with vitamin D deficiency during a weight reduction program. Moreover we wanted to analyse wheather the changes in body composition (FM, FFM) are better indicators of physical performance improvement than bmi in order to propose reliable monitoring practice during weight reduction programme.  

Conclusion of this paper is: Improving physical fitness is positively correlate with a decrease in FM (fat mass) and an increase in FFM(fat free mass) measured in both absolute values and %. Both BIA and DXA methods proved to be equally useful for measuring body composition. Since BIA and DXA have proved equally suitable for monitoring body composition, BIA seems to be a better method in clinical practice, because of its greater accessibility and lower onerousness. Intervention programs should focus not only on a direct reduction of BMI, but above all on improving body composition and lifestyle modification, because only such action gives a chance to preserve long-term effects in the form of preserving and im-proving health capital.

  1. The results of the analysis of the effects of vitamin D supplementation were discussed in the first work in the cycle that was cited. The results did not confirm the effect of supplementation on the improvement in weight reduction and no differences in body composition (fat mass, fat free mass measured by both BIA and DXA) between the group with vitamin D supplementation and the placebo group were found. Yes, endogenus vitamin D was considerd.
  2. Our intention was to present the resoults of our stydy in the contecst of other reserch. If we should  limit the dyscusion please suggest.
  3. FFM – fat free mass. All shortcuts have been expanded.

Round 2

Reviewer 2 Report

The manuscript has been improved and no further recommendation.